# Influence of Electrochemical Pretreatment Conditions of PtCu/C Alloy Electrocatalyst on Its Activity

**DOI:** 10.3390/nano11061499

**Published:** 2021-06-06

**Authors:** Angelina Pavlets, Anastasia Alekseenko, Vladislav Menshchikov, Sergey Belenov, Vadim Volochaev, Ilya Pankov, Olga Safronenko, Vladimir Guterman

**Affiliations:** 1Chemistry Faculty, Southern Federal University, 344090 Rostov-on-Don, Russia; angelina.pavlez@mail.ru (A.P.); an-an-alekseenko@yandex.ru (A.A.); men.vlad@mail.ru (V.M.); osafronenko@sfedu.ru (O.S.); gut57@mail.ru (V.G.); 2Institute of Physical and Organic Chemistry, Southern Federal University, 344090 Rostov-on-Don, Russia; vvolochaev@sfedu.ru (V.V.); ipankov@sfedu.ru (I.P.)

**Keywords:** platinum electrocatalyst, PtCu/C, oxygen reduction reaction, methanol electrooxidation, catalyst activity, de-alloyed catalysts, electrochemical activation

## Abstract

A carbon supported PtCu_x_/C catalyst, which demonstrates high activity in the oxygen electroreduction and methanol electrooxidation reactions in acidic media, has been obtained using a method of chemical reduction of Pt (IV) and Cu (2+) in the liquid phase. It has been found that the potential range of the preliminary voltammetric activation of the PtCu_x_/C catalyst has a significant effect on the de-alloyed material activity in the oxygen electroreduction reaction (ORR). High-resolution transmission electron microscopy (HRTEM) demonstrates that there are differences in the structures of the as-prepared material and the materials activated in different potential ranges. In this case, there is practically no difference in the composition of the PtCu_x-y_/C materials obtained after activation in different conditions. The main reason for the established effect, apparently, is the reorganized features of the bimetallic nanoparticles’ surface structure, which depend on the value of the limiting anodic potential in the activation process. The effect of the activation conditions on the catalyst’s activity in the methanol electrooxidation reaction is less pronounced.

## 1. Introduction

The increase in the catalytic activity of platinum-containing electrocatalysts is a topical problem for the commercial use of proton-exchange membrane fuel cells [1,2,3]. It is well known that platinum doping with some d-metals increases the catalyst activity due to a number of effects, as follows: (1) a change in the electronic structure of the metal; (2) a decrease in the interatomic distance in the metal crystal lattice, which contributes to the dissociative adsorption of oxygen molecules; (3) a change in the surface oxides’ composition and an increase in the corrosion resistance of the alloy in comparison to pure Pt [4,5,6]. One of the important points that is limiting the use of bimetallic catalysts is the leaching of the alloying component, which results in the poisoning of the proton-exchange membrane and the degradation of the membrane electrode assembly (MEA) [7,8]. Therefore, one of the current research areas of the present paper is the catalyst’s synthesis with nanoparticles of the following structures: core-shell [4,9,10,11], gradients [12,13], hollow [14,15,16], porous frameworks [17,18], nanowires, and nanodendrites [19,20,21]. The key components of such catalysts are nanoparticles with a platinum shell, which protects the d-metal core from dissolution during the fuel cell operation. Unfortunately, the formation of bimetallic nanoparticles with a uniform and continuous platinum shell on the carbon support surface is a very laborious and poorly scalable process.

Another approach is to obtain nanoparticles with a secondary core-shell architecture [22,23]. In this case, the bimetallic catalyst containing solid solution nanoparticles is treated, for example, with an acid. This leads to the removal of the alloying component atoms primarily from the NPs’ surface layers. As a result, a platinum shell is formed on the nanoparticles to protect the core-metal atoms from dissolution. The stability of the metal composition during such a catalyst operation in MEA increases. A reorganization of the nanoparticles’ structure resulting from the catalyst’s pretreatment in an aggressive environment affects their electrochemical performance [20,22,23,24].

The processes occurring with catalyst nanoparticles during electrochemical measurements are of great interest and has recently been actively presented in the literature [21,22,24,25,26,27,28]. The processes of bimetallic nanoparticles’ transformation during various current-forming reactions are also the subject of discussion [25]. Most publications consider the fundamental aspects of the model systems [29,30], in particular, with stabilized nanoparticle colloids being used [20,31], as well as model bimetallic systems obtained at high temperatures in a reducing atmosphere [25,32]. In works [23,24,25,26] it has been shown that electrochemical de-alloying makes it possible to obtain catalysts with a composition that remains stable in the process of further electrochemical measurements. Generally, such studies are carried out for the catalysts containing large nanoparticles of at least 10 nm. At the same time, complex laboratory synthesis methods are used to obtain catalysts that are not suitable for industrial use [6,24,25,26]. Wang et al. [27] compared the characteristics of the PtCu/C materials after chemical (acidic) and electrochemical de-alloying. It was noted that, after acid treatment, the activity of the catalysts in ORR decreased due to the dissolution of a part of the alloying component from nanoparticles, as well as a decrease in the average size of the nanoparticles.

The results of the above studies indicate that the conditions of the chemical or electrochemical de-alloying of PtM/C materials can play an important role in obtaining catalysts with the optimal composition/structure of bimetallic nanoparticles. We have previously shown that a de-alloyed PtCu/C catalyst can be successfully used in MEA after the acid treatment stage [33]. In our opinion, however, it is important to study the possibility of conducting electrochemical acid treatment and study the characteristics of the obtained de-alloyed PtCu/C under various conditions of such treatment. The aim of this study is to determine the effect of the voltammetric activation potential range on the composition and structure of the PtCu/C-alloy catalyst, and, as a consequence, on its activity in the reactions of oxygen electroreduction and methanol electrooxidation.

## 2. Materials and Methods

When obtaining the PtCu/C catalyst, Vulcan XC-72 (Cabot Corporation, Boston, MA, USA) with a surface area of 270 m^2^/g was used as a carbon support. Solutions of the precursors of platinum (H_2_PtCl_6_∙6H_2_O, JSC Aurat, Moscow, Russia) and copper (CuSO_4_∙5H_2_O, extra pure grade, JSC Vekton, Saint-Petersburg, Russia) were prepared using bidistilled water. An aqueous solution of sodium borohydride (NaBH_4_, JSC Vecton, Saint-Petersburg, Russia) was used as a reducing agent.

The synthesis of a PtCu/C sample containing bimetallic nanoparticles with an alloy structure (solid solution) was based on the simultaneous reduction of copper and platinum precursors from a solution of CuSO_4_∙5H_2_O and H_2_PtCl_6_∙6H_2_O. The reduction was carried out in a carbon suspension in a mixture of water–ethylene glycol at a ratio of 1:2 at a pH of 9–11. The reducing agent, a 0.5 M sodium borohydride solution, was taken in excess. The reaction mixture was kept under constant stirring for 60 min at room temperature. The resulting PtCu/C material was separated by filtration, washed repeatedly with isopropanol and bidistilled water, and then dried over P_2_O_5_.

The ratio of copper to platinum in the samples was determined using X-ray fluorescence (XRF) analysis on a spectrometer with total external reflection of X-ray radiation RFS-001 (Research Institute of Physics, SFedU, Rostov-on-Don, Russia). The sample exposure time was 300 sec. Registration and processing of X-ray fluorescence spectra was carried out using UniveRS software (SFedU). The content of the components was determined semi-quantitatively, relative to each other. For this, the integral areas under the peaks corresponding to Pt and Cu were calculated. Note that this method does not identify the state of the metal in the sample (atomic, ionic). The error of the XRF method in determining the composition was PtCu_x_ ± 0.02.

The mass fraction of metals in the sample was determined using gravimetry from the mass of the residue unburned when heated to 800 °C, assuming that it consists of Pt and copper (II) oxide and taking into account the atomic ratio of the metals determined using XRF analysis.

The PtCu/C catalyst was characterized by powder X-ray diffraction using a ARL X’TRA diffractometer (Thermo Scientific, Lausanne, Switzerland) with CuKα-radiation (λ = 0.154056 nm) at room temperature, and the diffraction patterns were collected in the 2-theta angle range of 15–55 degrees at a scan rate of 2° min^−1^. Fitting of the X-ray diffraction pattern was performed in the SciDAVis program using the Lorentz function; the results from the approximation and separation of contributions of different reflections were used in further calculations. The average crystallite size of the metal phase was calculated for the most intense peak (111) using the following Scherrer equation: D = Kλ/(FWHMcosθ), where λ is the wavelength of monochromatic radiation (in Å), FWHM is the half-width of the reflection at half of the maximum (in radians), K = 0.89 is the Scherrer constant, D is the average thickness of the “stack” of reflecting planes in the coherent scattering region (in Å), and θ is the angle of incidence of the X-ray beam (in radians). The accuracy of D_av_ determination was ±5%. More detailed information is described in [34].

The analysis of the size distribution, spatial distribution, and morphological types of NPs was performed using HRTEM (JEOL JEM-F200, with an accelerating voltage of 200kV, a cold field emission electron gun, and a double-tilt Be specimen holder 01361 RSTHB, CMOS AMT camera). The resolution range varied from 100 to 1 nm. The typical exposure time was 500 ms. The instrument was equipped with a Bruker QUANTAX EDS system, including an energy-dispersive Peltier-cooled XFlash detector and ESPRIT 2 software package. For element 2D mapping and line scan procedures by EDS, a STEM mode was used with identical settings and with an exposure time from 180 to 500 s.

To prepare a sample for measurements, 0.5 mg of the catalyst was placed in 1 mL of isopropanol and dispersed by ultrasound for 10 min. A drop of the resulting suspension was applied to a standard copper mesh with a diameter of 3.05 mm, covered with a 5–6-nanometer thick layer of amorphous carbon, and next the sample was dried in air at room temperature for 60 min. The histograms of the platinum nanoparticles size distribution in the catalysts were plotted due to the results of determining the sizes of at least 400 randomly selected particles in the TEM images in different regions of the sample.

Catalytic ink was used to prepare the test electrode. When preparing the ink, a 6-milligram sample of the catalyst was added to a mixture of 900 μL of isopropanol and 100 μL of a 0.5% aqueous–alcoholic emulsion of the Nafion^®^ polymer. Then, the suspension was stirred with a magnetic stirrer for 5 min and subjected to dispersion in an ultrasonic bath for 10 min, with the water temperature being controlled, as described in [34]. The described stirring and sonication procedure was repeated twice. An aliquot of ink with a volume of 6 μL was applied to the polished and degreased glassy carbon of a rotating disk electrode (RDE) with an area of 0.196 cm^2^, the weight of the drop was recorded. To increase the uniformity of the catalytic layer, the deposited drop was dried in air for 5 min by rotating the electrode at 700 rpm (Pine Research Instrumentation, Inc.).

The electrochemical behavior of the catalysts was studied in a standard three-electrode cell using cyclic voltammetry at a temperature of 23 °C on a potentiostat VersaSTAT 3 (Princeton Applied Research, Oak Ridge, Tennessee, USA). A silver chloride electrode was used as a reference electrode, and a platinum wire was used as a counter one. All of the potentials in this work are given relative to a reversible hydrogen electrode (RHE). Before carrying out electrochemical measurements, the electrode was subjected to electrochemical activation by setting 100 cycles of voltammetry in the potential range of 0.04–1.20 V or 0.04–1.0 V, at a scanning rate of 200 mV/s, in a 0.1 M HClO_4_ solution, and in an Ar atmosphere. After the stage of standardization (activation), the electrolyte was replaced in the cell. Then, two CVs were recorded in the potential range of 0.04–1.20 V or 0.04–1.0 V, respectively, with a scanning rate of 20 mV/s for further calculation of the electrochemically active surface area (ESA) value from the amount of electricity consumed for hydrogen adsorption/desorption [34].

A measurement of the catalyst’s activity in the oxygen reduction reaction (ORR) was carried out based on the analysis of voltammograms recorded with different rotating speeds of RDE in an O_2_ saturated electrolyte. To achieve this, a recording of voltammograms with a linear potential scan towards high values was performed at a rate of 20 mV/s at the following four rotation speeds of the disk electrode: 400, 900, 1600, and 2500 rpm. The contribution of the ohmic voltage drop was taken into account according to the following formula: *E* = *E*_set_ – I*R, where *E*_set_ was the set value of the potential and I*R was the ohmic potential drop equal to the product of the current strength and the resistance of the solution layer between the reference electrode and the investigated electrode, which was 23 ohms. To calculate the contribution of the processes occurring at the electrode in the deoxygenated solution (Ar atmosphere), a similar curve, recorded at the same electrode during measurements in an Ar atmosphere, was subtracted from the voltammogram as follows: I(O_2_)-I(Ar). The catalyst’s activity in the ORR (kinetic current) was determined from the normalized voltammograms, with the contribution of mass transfer under the conditions of using RDE being considered. The calculation of the kinetic current at a potential of 0.90 V (RHE) was carried out according to the following Koutecký–Levich equation: 1/j = 1/j_k_ + 1/j_d_, where j was the experimentally measured current, j_d_ was the diffusion current, and j_k_ was the kinetic current. The mass and specific activities were calculated by dividing the kinetic current by the mass of platinum on the electrode and by the determined ESA, respectively.

While studying the catalyst’s activity in MOR, methanol and perchloric acid were added to the electrochemical cell, thereby obtaining a solution of 0.1 M HClO_4_ + 0.5 M CH_3_OH in an argon atmosphere. Cyclic voltammograms were recorded in the potential range of 0.04–1.3 V with a potential sweep rate of 20 mV/s. Chronoamperograms were measured at a potential of 0.87 V. The measurements were carried out in an argon atmosphere.

A commercial Pt/C (HiSPEC3000, JohnsonMatthey, JohnsonMatthey PLC, London, Great Britain) analogue with a mass fraction of platinum equal to 20%, marked JM20, was used as a conventional catalyst.

After electrochemical measurements, the PtCu/C catalysts were removed from the end of the glassy carbon electrode, then the ratio of metals in the samples was measured using the X-ray fluorescence method, as described above. To study the microstructure of the PtCu/C catalysts after the electrochemical activation stage, additional experiments were carried out on a graphite plate with an area of 1.805 cm^2^ to obtain a sufficient amount of the examined material.

## 3. Results

The composition of the obtained PtCu/C catalyst according to XRF data was PtCu_1.3_/C (Table 1) and the metal content according to XRF and gravimetry data was 7.8 wt% Cu and 17.9 wt% Pt. These results are in good agreement with the values calculated from the precursor loading.

It has been proven using X-ray diffraction (Figure 1) that the studied PtCu/C and commercial Pt/C materials contain reflections of the carbon and platinum (face-centered cubic lattice, space group Fm3m) phase. The XRD pattern of the Pt/C analog demonstrates (111) and (200) reflections of pure platinum. Note that for the bimetallic catalysts (111) and (200), the reflections of platinum are shifted to the region of larger 2-theta angles, compared to the pure Pt, which is due to the formation of a Pt–Cu solid solution [34,35]. From the position of the reflections, one can estimate the crystal lattice parameter (a) of the Pt–Cu solid solution and estimate its composition according to Vegard’s law [36,37]. For the obtained PtCu/C catalyst, the calculated a value is 3.815 Å and the bimetallic nanoparticles composition is PtCu_0.54_. The average crystallite size calculated using the Sheerer equation is 2.7 ± 0.1 nm and 2.5 ± 0.1 nm for PtCu/C and JM20, respectively.

The diffractogram of PtCu/C in Figure 1 does not show the peaks corresponding to the phase of Cu and its oxides. [38]. At the same time, this fact does not mean the absence of copper oxides since these oxides can be present in the form of amorphized structures [39]. The discrepancy between the sample composition using XRF and the Pt–Cu solid solution determined according to Vegard’s law can be explained by the incomplete incorporation of copper into the Pt–Cu solid solution. Some amount of copper is apparently contained in the catalyst in the form of amorphous copper oxide inclusions, as was shown in [39]. In addition, taking into account the high probability of an uneven distribution of platinum and copper atoms in the nanoparticles, it can be assumed that the inhomogeneity of the particle structure contributes to the broadening of the maxima in X-ray diffractograms, which leads to a certain underestimation of the crystallite size calculated using the Scherrer equation [13,34].

The presence of metal nanoparticles on the carbon support surface in the obtained PtCu/C material was also confirmed by the TEM study results (Figure 2a,b). The spherical nanoparticles, ranging in size from 3 to 5 nanometers, were evenly distributed over the carbon support surface (Figure 2). At the same time, there were particle agglomerates with a size of 9–10 nm (Figure 2a–c). Based on the analysis of nanoparticles sizes in PtCu/C material, a histogram of the nanoparticles’ size distribution was constructed (Figure 2c). Note that the average nanoparticle size determined from the TEM was about 5 nm, which significantly exceeded the value of the average crystallite diameter (Figure 2c) according to the XRD data. It is known that the average size of crystallites determined using XRD is usually smaller than the size of nanoparticles determined using TEM [40,41]. The main reasons for this difference are as follows: one nanoparticle can consist of several crystallites; the presence of an amorphous layer on the nanoparticle surface that is not detected using XRD; and there are different approaches to calculating the average crystallite size [42].

The line scanning of the elemental composition of PtCu/C material (Figure 2d,e) confirms the uniform distribution of components in PtCu nanoparticles. Elemental mapping of a separate fragment of PtCu/C material also confirms the coincidence of the localization of copper and platinum (Figure 3). This means that each nanoparticle in the studied fragment contains both platinum and copper atoms. Thus, the results of the transmission electron microscopy, elemental mapping of the surface, and analysis of the individual nanoparticles composition confirm that the studied PtCu/C material predominantly contains bimetallic Pt–Cu nanoparticles with an alloy structure.

The stage of electrochemical activation is the initial stage of all of the electrochemical measurements for Pt-based catalysts [43,44,45,46]. It is obvious that some atoms of the alloying component in the PtCu/C material are located on the surface of bimetallic nanoparticles. In addition, some of the copper atoms can be included in amorphous oxides. All of these atoms can dissolve because of interaction with electrolyte components. As a result, during the activation of PtCu/C catalysts, not only is the surface of the nanoparticles cleaned of existing contaminants, but also the composition/structure of their surface changes due to the predominant dissolution of the alloying component [26,27,35]. In fact, the stage of the electrochemical activation combines the acid treatment of catalysts with the electrochemical de-alloying of bimetallic nanoparticles.

In accordance with the purpose of the study stated above, a PtCu/C electrode investigation after voltammetric activation in two different modes was carried out. The first was activated in the potential range of 0.04–1.0 V and the second, in a wider range, from 0.04 to 1.2 V. Note that researchers used both activation modes [20,27,35], and sometimes expanded the range of potentials, for example, from 0.05 to 1.35 V [26].

The CV curves measured during the electrochemical activation of the PtCu/C catalyst (Figure 4a) do not show clearly pronounced hydrogen adsorption/desorption peaks in the potential range of 0.04–0.35 V, which is typical of bimetallic catalysts [27]. In addition, when cycling the bimetallic catalyst in the potential range of 0.04–1.2 V, anodic peaks near the 0.3 V potential are present on the first CVs (Figure 4a, black curve). Their appearance is due to the dissolution of copper from its own phase [27,47,48]. During PtCu/C material cycling, the anodic peaks of copper dissolution disappear, and the peaks of hydrogen adsorption/desorption appear more clearly. This fact indicates the enrichment of the nanoparticles surface with platinum [27].

The cyclic voltammograms of the PtCu/C sample recorded after the activation stage (Figure 4b,c) have a form typical of platinum-containing catalysts supported on Vulcan XC-72 [11,27,49]. In the hydrogen and double-layer regions, the form of CVs obtained for both for PtCu/C and for conventional Pt/C electrodes, as well as the values of the currents, do not actually depend on the activation conditions used (Figure 4c,d). This means that the ESA value of the activated catalyst is independent of the activation conditions. The ESA values of de-alloyed PtCu/C samples and Pt/C catalysts are about 40 m^2^/g(Pt) and 75–80 m^2^/g(Pt), respectively (Table 1). The compositions of the de-alloyed materials after the activation stage are also very close to each other (Table 1).

Despite the similar compositions and ESA values, bimetallic electrodes activated under different conditions exhibit different ORR activity: the sample activated in the potential range of 0.04–1.0 V demonstrates a mass activity two times higher than the sample activated in a potential range from 0.04 to 1.2 V (Table 1, Figure 5a,b). According to the results of calculations based on the Koutecký–Levich equation [50,51], at a potential of 0.90 V, the PtCu/C catalyst (1.0 V) is characterized by a kinetic current of 5.2 mA, while for the PtCu/C catalyst (1.2 V) it is of 2.8 mA only. It is important that the change of the activation conditions does not affect the ORR activity of the commercial Pt/C analog (Table 1, Figure 5c,d). The kinetic current, about 1.3 mA, is the same for catalysts JM20 (1.0 V) and JM20 (1.2 V) (Figure 5f). For all of the investigated catalysts, the number of electrons participating in the ORR at E = 0.9 V is close to four (Table 1). Nevertheless, the PtCu/C catalyst activated at the narrower potential range demonstrates a higher value of the half-wave potential in the ORR, which also indicates its higher activity compared to PtCu/C (1.2 V) (Table 1, Figure 5a,b). The considerable difference in the ORR activity values for the PtCu/C (1.0 V) and PtCu/C (1.2 V) samples might indicate major differences in the structure and composition of the PtCu nanoparticles’ surface layers after catalyst activation (Table 1). Taking into account the independence of the composition and ESA of PtCu/C catalysts in the activation conditions, it can be assumed that the nature of the structural reorganization of bimetallic nanoparticles, which accompanies the partial removal of base metal atoms, depends on the value of the limiting positive potential. Similar effects have been observed by Wang et al. [27] in a study of the changes in the structure of Cu3Pt nanoparticles during an acid treatment and an electrochemical de-alloying.

To study the changes in the material microstructure during electrochemical activation, a TEM study of PtCu/C (1.0 V) and PtCu/C (1.2) samples extracted from the electrochemical cell after the completion of activation was carried out (Figure 6). In contrast to the “as-prepared” catalyst, the TEM photographs of the materials after electrochemical activation show the NPs agglomerates in the form of rods 7–8 nm in length and 3–3.5 nm in width (Figure 6a,b,d,e). Unfortunately, only the separately located spherical nanoparticles were considered when constructing histograms of the particle size distribution of the samples (Figure 6c,f,).

Apparently, after the activation of the PtCu/C catalyst in the potential range of 0.04–1.0 V the average nanoparticle size decreased from 5 to 4 nanometers (Figure 2c and Figure 6c) due to the selective dissolution of the copper atoms from the NPs’ surface layers. In this case, the high activity of the catalyst in the ORR could be due to several factors. Among them are defects, which appear in the structure of the nanoparticles; and the preservation of the positive effect of the alloying component atoms, which are protected by a thin platinum shell. By expanding the potential range of activation to 1.2 V, the de-alloying of copper leads to the formation of a less efficient nanoparticle structure. Interestingly, in this case, the size of the nanoparticles remained practically unchanged (Figure 6f). The line scanning of the elemental composition of PtCu/C material after activation (Figure 7) also confirms the uniform distribution of components in PtCu nanoparticles as in the “as-prepared” material. Unfortunately, this method did not reveal differences in the microstructure of PtCu nanoparticles after electrochemical activation under different conditions. We assume that a part of the PtCu/C nanoparticles of the sample after activation to 1.2 V also decreases in size due to the selective dissolution of copper, as in the case of activation to 1.0 V. At the same time, the NPs’ agglomeration intensified, leading to the formation of larger particles up to 11 nm (Figure 6f). That leads to the preservation of the same average particle size as in the “as-prepared” sample. Thus, the average size of nanoparticles does not undergo significant changes as a result of activation, but the nature of the particles size distribution does change. Note that in some works, the effect of a decrease in the size of bimetallic nanoparticles during the acid treatment of catalysts was described. The authors of [24,25] associated this effect with surface reorganization and the appearance of structural defects that appeared and positively affected the activity in the ORR.

Further, the behavior of the catalysts obtained after activation under various conditions was studied in the methanol electrooxidation reaction (MOR). Usually, when comparing the activity of platinum-containing catalysts in the alcohols’ oxidation using voltammetry, several indicators are used, as follows: the onset potential (E_onset_) [52,53,54]; the specific current value at the selected potential [53]; the maximum specific oxidation current (Imax) at the forward sweep of the potential [55,56]; as well as the amount of electricity consumed for the oxidation of alcohol (Q), which is calculated in the forward sweep of the potential [33]. The method of chronopotentiometry allows one to study the change in current value over time; usually, the values of the initial current (I_initial_) and the current after experiment (I_final_) are compared. A decrease in its value is due to the gradual poisoning of the catalyst by intermediate products of alcohol electrooxidation [33,57]. Therefore, the analysis of the change in the specific current with time makes it possible to assess the catalyst’s tolerance to intermediates.

A comparative study of the activated PtCu/C catalysts by CV (Figure 8a) showed that despite the lower ESA values compared to the commercial Pt/C (JM20) catalyst (Table 1), their mass activity in MOR is two or three times higher (Table 2). This follows from a comparison of the values of both Q_CH3OH_ and Imax. At the same time, the onset potential values for PtCu/C and Pt/C (JM20) catalysts do not actually differ (Figure 8a).

According to the results of chronoamperometry, the PtCu/C catalyst also shows greater activity in MOR and a satisfactory stability to intermediate oxidation products compared to the Pt/C commercial material (Figure 8b). For PtCu/C catalysts, the decrease in specific current during the measurements was 44–45%, and for Pt/C commercial material it was 61% of the initial value (Figure 8b, Table 2).

At the same time, in contrast to the ORR, the significant influence of the potential range during activation of PtCu/C materials on the parameters of CV and chronoamperometry, and, therefore, on the catalyst’s activity in the MOR has not been established (Figure 8a). Samples activated up to 1.0 and up to 1.2 V showed close values of E_onset_, Q_CH3OH_, and I_max_ (Table 2). Apparently, the role of the changes in the structure of bimetallic nanoparticles, which occur during activation, is large in the case of the ORR and insignificant for the MOR.

## 4. Conclusions

A PtCu_1.3_/C-alloy material containing bimetallic nanoparticles with a “solid solution” structure was obtained using the wet-synthesis method under conditions of simultaneous reduction of copper and platinum precursors.

It was found that the value of the potential range, or rather the value of the maximum anodic activation potential, had a significant effect on the activity of electrochemically de-alloyed PtCu_0.25_/C electrocatalysts in the ORR. The voltammetric activation mode of a PtCu_1.3_/C-alloy catalyst in the potential range from 0.04 to 1.0 V made it possible to obtain a sample with an activity in ORR about two times higher than that of a sample activated in the potential range from 0.04 to 1.2 V. In this case, the composition of the metal component of the de-alloyed catalysts and the value of their ESA did not depend on the activation conditions.

The electrochemical activation of the PtCu_1.3_/C catalyst in the potential range from 0.04 to 1.0 V led to an insignificant decrease in the average size of nanoparticles, which might be due to the selective dissolution of copper atoms from the surface layers of nanoparticles. The average size of the PtCu nanoparticles in the catalyst obtained after activation in the potential range from 0.04 to 1.2 V practically did not change, since, in this case, the formation of larger nanoparticles was enhanced due to the aggregation of smaller ones. No significant effect of the platinum–copper catalyst activation mode on its electrochemical behavior in MOR was found.

Taking into account the literature’s data [21,22,25,26,27,28], the established effect of activation conditions on the activity of de-alloyed platinum–copper catalysts in the ORR may be due to major differences in the structure (defectiveness) and composition of the surface layers of PtCu nanoparticles. Unfortunately, it is not possible to obtain direct confirmation of such differences within the framework of this work. Such an attempt will be carried out in the next stage of our research.

In general, bimetallic de-alloyed electrocatalysts are significantly more active than the commercial Pt/C analog, both in the ORR and MOR. The study showed that changes in the conditions of the electrochemical activation of bimetallic electrocatalysts can be used as an effective way to increase their catalytic activity in some electrochemical reactions. Therefore, the authors recommend using protocols of electrochemical measurements with a range of activation potentials up to 1.0 V, when studying electrocatalysts based on bimetallic nanoparticles.

## Figures and Tables

**Figure 1 nanomaterials-11-01499-f001:**
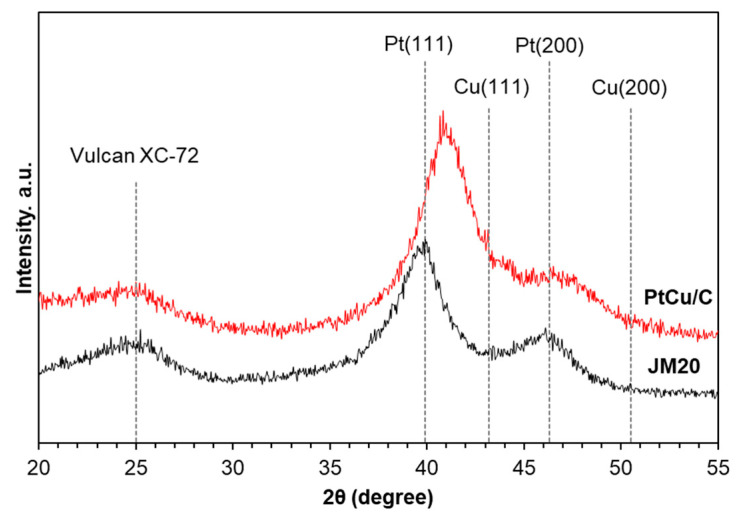
X-ray diffraction patterns of the “as-prepared” PtCu/C and Pt/C (JM20) materials (the vertical lines indicate the values of the characteristic reflections of pure carbon support, platinum, and copper).

**Figure 2 nanomaterials-11-01499-f002:**
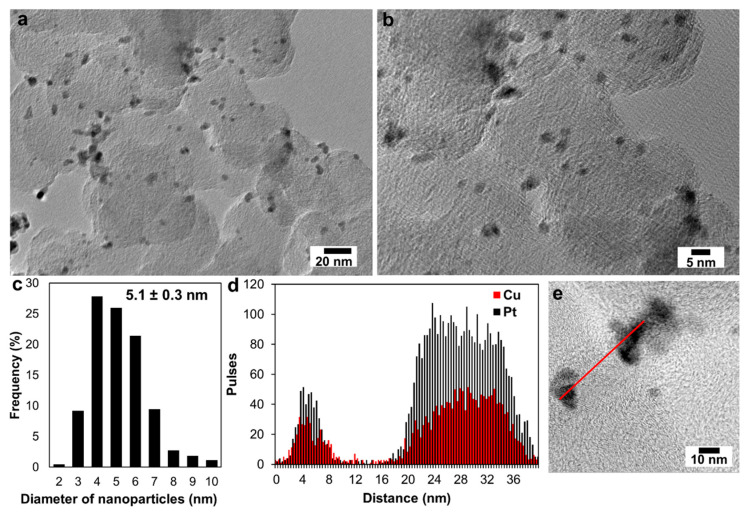
Transmission electron microscopy photographs of a PtCu/C sample (**a**,**b**) and histograms of the nanoparticle size distribution in the corresponding material (**c**). Line profiles of Pt and Cu (**d**) obtained by scanning nanoparticles (**e**).

**Figure 3 nanomaterials-11-01499-f003:**
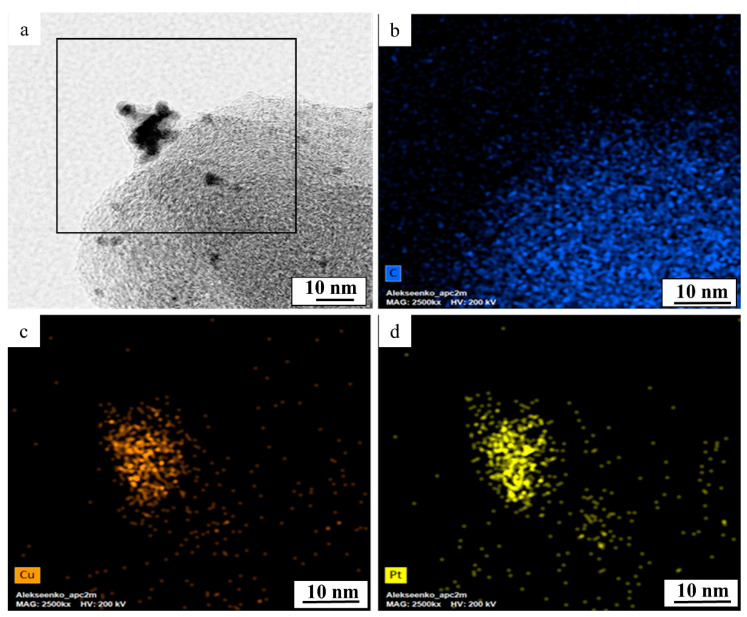
EDX mapping of a PtCu/C sample. TEM image of a catalyst fragment (**a**) and maps of the distribution of the following analyzed elements: carbon (**b**), copper (**c**), and platinum (**d**).

**Figure 4 nanomaterials-11-01499-f004:**
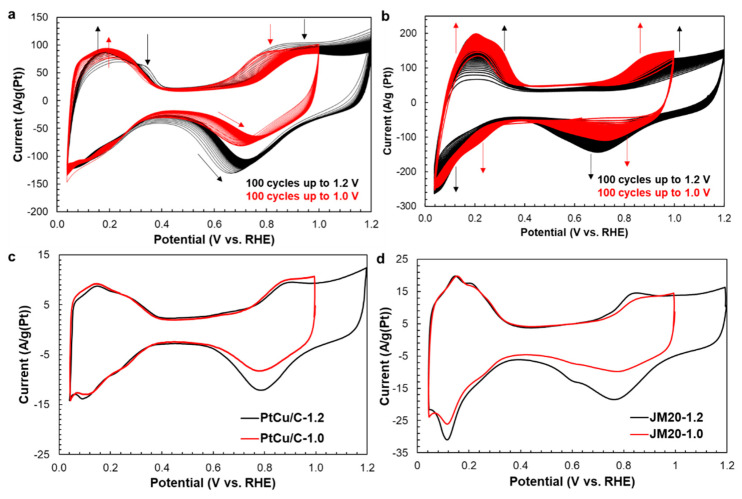
CV of PtCu/C (**a**,**c**) and Pt/C (**b**,**d**) catalysts at the activated stage (**a**,**b**) 100 CVs, potential scan rate is 100 mV/s and after activation stage (**c**,**d**) 2 CVs, potential scan rate is 20 mV/s. The electrolyte is a 0.1 M HClO4 solution saturated with argon.

**Figure 5 nanomaterials-11-01499-f005:**
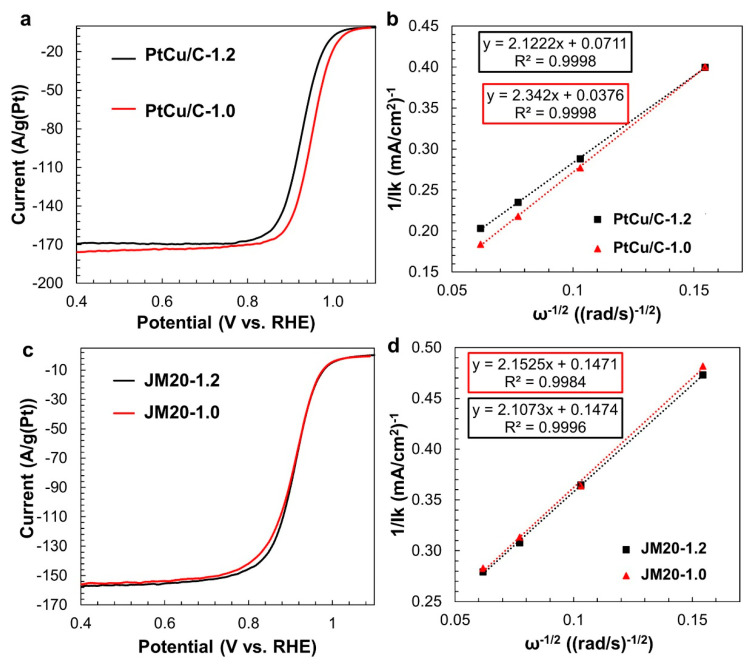
LSV curve for ORR (**a**,**c**) and the corresponding dependences 1/I_k_ vs ω^−1/2^ at a potential of 0.90 V (**b**,**d**). Rotation speed is 1600 rpm and potential sweep rate is 20 mV/s. The electrolyte is a 0.1 M HClO_4_ solution saturated with O_2_.

**Figure 6 nanomaterials-11-01499-f006:**
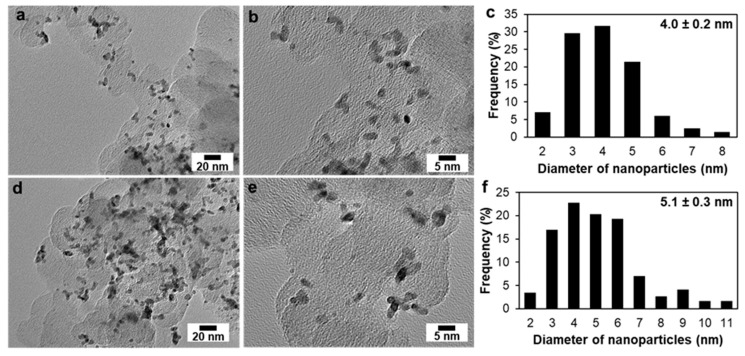
Transmission electron microscopy photographs of PtCu/C-1.0 (**a**,**b**) and PtCu/C-1.2 B (**d**,**e**). Samples and histograms of the nanoparticle size distribution in the corresponding material PtCu/C-1.0 (**c**) and PtCu/C-1.2 (**f**).

**Figure 7 nanomaterials-11-01499-f007:**
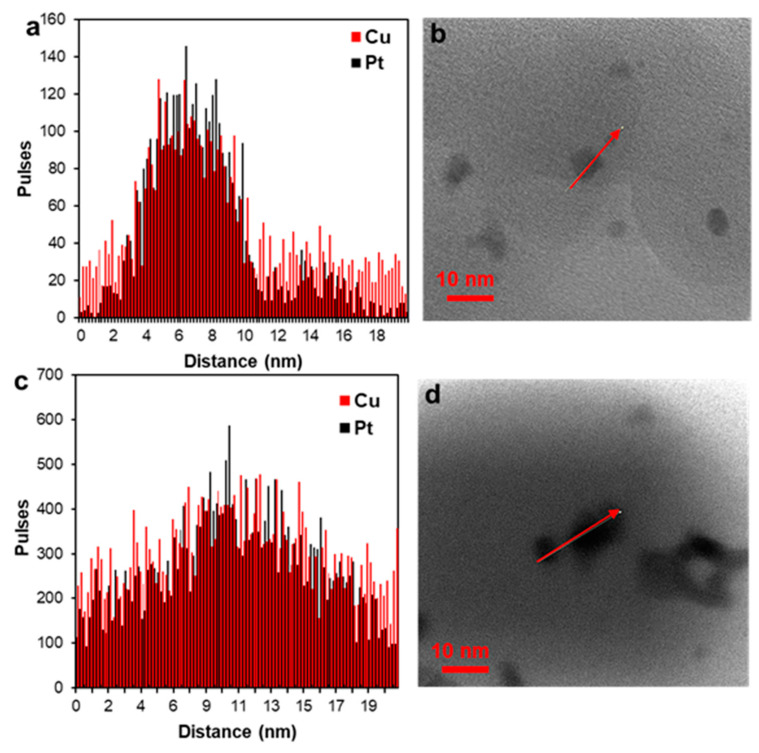
Line profiles of Pt and Cu (**a**,**c**) obtained by scanning nanoparticles (**b**,**d**). Samples: PtCu/C-1.0 (**a**,**b**); PtCu/C-1.2 (**c**,**d**).

**Figure 8 nanomaterials-11-01499-f008:**
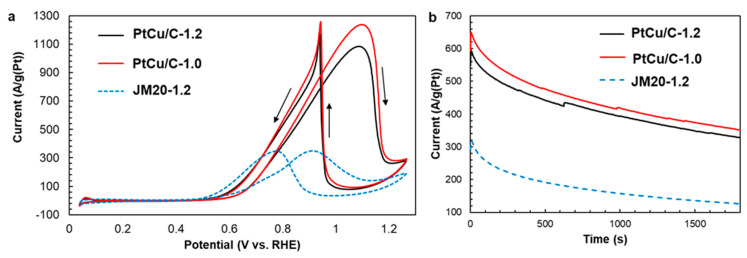
CV (**a**) and chronoamperograms (**b**) of PtCu/C catalysts obtained after activation under different potential ranges. Currents are normalized to the mass of platinum. The potential sweep rate is 20 mV/s. The electrolyte is a 0.1 M HClO_4_ + 0.5 M CH_3_OH solution saturated with Ar at atmospheric pressure.

**Table 1 nanomaterials-11-01499-t001:** Composition and some electrochemical parameters of PtCu/C and Pt/C catalysts after the activation in different potential ranges.

Sample	Initial Composition	Potential Range at the Activation Stage, V	Composition after the Activation Stage	ESA, m^2^g^−1^ (Pt)	I_k_, Ag^−1^ (Pt)	I_k_, Am^−2^ (Pt)	Number of ē	E_1/2_, V
**PtCu/C**	PtCu_1.3_	0.04–1.20	PtCu_0.25_	41	417	10.1	4.3	0.93
		0.04–1.00	PtCu_0.24_	43	878	20.3	3.9	0.95
**JM20**	Pt	0.04–1.20	-	80	181	2.2	4.3	0.92
		0.04–1.00	-	75	184	2.2	4.2	0.93

**Table 2 nanomaterials-11-01499-t002:** Parameters characterizing the behavior of PtCu/C and Pt/C catalysts in the methanol electrooxidation reaction.

Sample	Potential Range at the Activation Stage, V	Q_CH3OH_, C/g(Pt)	I_max_, Ag^−1^(Pt)	E_onset_, V	I_initial_, Ag^−1^(Pt)	I_final_, Ag^−1^(Pt)
PtCu/C	0.04–1.00	18,703	1236	0.51	650	352
PtCu/C (1.2 V)	0.04–1.20	16,647	1084	0.50	596	328
JM20	0.04–1.20	5648	350	0.51	320	127

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
