# Peer review of "Influence of Electrochemical Pretreatment Conditions of PtCu/C Alloy Electrocatalyst on Its Activity"

_nanomaterials, 2021, doi:10.3390/nano11061499_

Round 1

Reviewer 1 Report

Dear Authors, I found this work readable, well arranged, offering good set of experimental results, thus very interesting. Nevertheless I cannot recommend this in the present form: (i) you should underline the novelty, there are papers describing PtCu/C Alloys in the literature, (ii) you should show some more information about C-support;

Detailed questions:

How HRTEM analysis was performed?

I found the XRD pattern (Fig 1) bit strange. Can you please discuss deeper your results and join some additional experimental details?

Chapter 4 is in fact the “summary” rather than “Conclusions”, can you correct it?

Author Response

The authors express their gratitude to the respected reviewers for carefully reading the manuscript, comments and recommendations made! The text of the article has been revised and supplemented. The most significant additions have been made to the "Materials and Methods" section. The changes made are marked with a yellow marker. Below are the answers to the questions and comments of the reviewer.

Dear Authors, I found this work readable, well arranged, offering good set of experimental results, thus very interesting. Nevertheless I cannot recommend this in the present form:

(i) you should underline the novelty, there are papers describing PtCu/C Alloys in the literature,

(ii) you should show some more information about C-support.

We agree with the reviewer that there are many publications demonstrating the positive effect of alloying platinum with copper on catalyst activity. Among them are our articles:

A.S. Pavlets, etc. A novel strategy for the synthesis of Pt-Cu uneven nanoparticles as an efficient electrocatalyst toward oxygen reduction, Int J Hydr Energy, 2021, 46(7), 5355-5368;

  1. Menshchikov, etc. Effective Platinum-Copper Catalysts for Methanol Oxidation and Oxygen Reduction in Proton-Exchange Membrane Fuel Cell, Nanomaterials 2020, 10(4) 742;

A.A. Alekseenko, etc. Pt/C electrocatalysts based on the nanoparticles with the gradient structure, Int. J. of Hydrogen Energy, 2018, 43(7), 3676 – 3687;

A.A. Alekseenko, etc. Durability of de-alloyed PtCu/C electrocatalysts, Int. J. of Hydrogen Energy, 2018, 43(51), P. 22885-22895;

V.V. Pryadchenko, etc. Effect of Thermal Treatment on the Atomic Structure and Electrochemical Characteristics of Bimetallic PtCu Core-Shell Nanoparticles in PtCu/C Electrocatalysts, J. Phys. Chem. C, 2018, 122(30), 17199–17210, etc.

The novelty of this article lies in the fact that, using the example of PtCu/C-alloy catalyst, for the first time, a significant effect of the conditions of the electrochemical activation, preceding ANY electrochemical measurements, on the ORR activity of bimetallic catalysts is shown. We have underline the novelty in the revised Abstract, and have added additional explanations to highlight the novelty in other sections of the article. All additions and changes in the article are highlighted in yellow.

Regarding carbon support information, the "Materials and Methods" section indicates that carbon black was used in the synthesis of the catalysts (Cabot Corp.). The same support is used by the manufacturers of the Pt/C catalyst HiSPEC3000, which we studied as a "conventional" catalyst. Vulcan XC-72 is well known in the art, widespread and most commonly used as a support for platinum catalysts. Its structural and morphological characteristics have been described many times in the literature. Our study does not consider the processes caused by the influence of the carbon support nature. Therefore, we only added information about the specific surface area of ​​this support in the "Materials and Methods" section.

Detailed questions:

How HRTEM analysis was performed?

A detailed description of the HRTEM measurement technique has been added to the experimental part.

I found the XRD pattern (Fig 1) bit strange. Can you please discuss deeper your results and join some additional experimental details?

We agree with the comment of the reviewer. An X-ray diffraction pattern of a commercial Pt/C catalyst is added to Fig. 1. Reflections typical of pure platinum are observed for the Pt/C sample. Taking into account the remarks, a more detailed description of the XRD study results has been carried out.

Chapter 4 is in fact the “summary” rather than “Conclusions”, can you correct it?

Thank you for your comment! Amendments have been made to the Conclusions section. However, we would like to point out that the "Results and Discussion" section contains quite a lot of information explaining the observed effects. In order not to repeat these explanations, we preferred to keep the "Conclusions" section compact, introducing only the main conclusions in a short form.

Reviewer 2 Report

The manuscript present results on electrochemical activity of PtCu/C alloy. Unfortunately it is not well enough written as to put in value the obtained results and reach the reader. It needs elaborate editing improvement before further consideration for publication.   

Author Response

The authors express their gratitude to the respected reviewers for carefully reading the manuscript, comments and recommendations made! The text of the article has been revised and supplemented. The most significant additions have been made to the "Materials and Methods" section. The changes made are marked with a yellow marker. Below are the answers to the questions and comments of the reviewer.

The manuscript present results on electrochemical activity of PtCu/C alloy. Unfortunately it is not well enough written as to put in value the obtained results and reach the reader. It needs elaborate editing improvement before further consideration for publication.   

Thanks for the comment, the text of the article has been supplemented and edited. The quality of the English language has been improved.

Reviewer 3 Report

In this manuscript, the authors demonstrated that the value of the potential range has a significant effect on the activity of the resulting de-alloyed PtCu0.25/C in the ORR. The catalyst activated in the potential range of 0.04 to 1.0V will have higher catalytic activity. However, the catalyst after activation has not been clearly characterized to confirm the difference between the catalyst before and after catalysis. Overall, the research is interesting. I suggest a minor revision before publication:

  • Where is Figure 5e,f mentioned in lines 267 and 268?
  • Why is the half-wave potential of the catalyst activated between 0.04-1.0V shifted positively, but the number of transferred electrons decreased?

Author Response

The authors express their gratitude to the respected reviewers for carefully reading the manuscript, comments and recommendations made! The text of the article has been revised and supplemented. The most significant additions have been made to the "Materials and Methods" section. The changes made are marked with a yellow marker. Below are the answers to the questions and comments of the reviewer.

Where is Figure 5e,f mentioned in lines 267 and 268?

Thank you for your comment. The error has been fixed.

 Why is the half-wave potential of the catalyst activated between 0.04-1.0V shifted positively, but the number of transferred electrons decreased?

 The shift in the half-wave potential is a consequence of the acceleration of the reaction, and the number of electrons is associated with the ORR mechanism. In our work, as in many publications, the accuracy of determining n is no more than ± 0.2. Therefore, all the n values presented in Table 1 confirm the 4-electron mechanism of oxygen electroreduction, and the values 3.9 and 4.3, in fact, correspond to n = 4.

Reviewer 4 Report

The authors demonstrated improve activity of platinum electrocatalyst by alloying with copper (PtCu/C) for oxygen reduction reaction (ORR) and methanol oxidation reaction (MOR). Due to the surface reorganization during cyclic voltametric activation, PtCu/C exhibited different ORR activity before and after activation process. In terms of ORR activity, activated PtCu/C shows higher ORR activity than commercial Pt/C. The authors also investigated MOR activity of PtCu/C. In MOR, PtCu/C shows about 4 times higher current than Pt/C and exhibits high stability in MOR. I recommend the acceptance of the manuscript if it is properly revised addressing the following issues:
1. In Figure 5, maximum currents (Ik) of PtCu/C and Pt/C are around -170 A/g and -150 A/g, respectively. However, in Table 1, the listed Ik values are 417 (for 1.2 V of PtCu/C), 878 (for 1.0 V of PtCu/C) and around 180 (for Pt/C). Please explain why values are different and which value is correct.

  1. Please analyze ORR stability in terms of chronoamperometry method. It is helpful for authors to confirm both ORR and MOR stability of PtCu/C.
  2. Please explain why the particle size of PtCu/C is increased after 1.2V activation. Also, PtCu/C (1.2V) exhibits low ORR current compared with PtCu/C (1.0V). Please explain the reason and advantage of activation in potential range up to 1.2 V vs RHE in terms of “solid solution” structure.
  3. In Table 1, electron transfer number is higher than 4. I think that in error. Please analyze electron transfer number by using rotating ring disk electrode (RRDE) in order to calculate correct number.
  4. In Table 1, cyclic voltametric activation up to 1.2 V vs RHE is much harsher condition than that of the 1.0 V. However, after activation, 1.0 V sample has 0.24 of Cu amount while 1.2 V sample has 0.25 of Cu amount. Please explain why 1.2 V sample has much copper amount.

6. We suggests that authors analyze BET surface area and compare with the ECSA results.
Author Response

The authors express their gratitude to the respected reviewers for carefully reading the manuscript, comments and recommendations made! The text of the article has been revised and supplemented. The most significant additions have been made to the "Materials and Methods" section. The changes made are marked with a yellow marker. Below are the answers to the questions and comments of the reviewer.

The authors demonstrated improve activity of platinum electrocatalyst by alloying with copper (PtCu/C) for oxygen reduction reaction (ORR) and methanol oxidation reaction (MOR). Due to the surface reorganization during cyclic voltametric activation, PtCu/C exhibited different ORR activity before and after activation process. In terms of ORR activity, activated PtCu/C shows higher ORR activity than commercial Pt/C. The authors also investigated MOR activity of PtCu/C. In MOR, PtCu/C shows about 4 times higher current than Pt/C and exhibits high stability in MOR. I recommend the acceptance of the manuscript if it is properly revised addressing the following issues:

  1. In Figure 5, maximum currents (Ik) of PtCu/C and Pt/C are around -170 A/g and -150 A/g, respectively. However, in Table 1, the listed Ikvalues are 417 (for 1.2 V of PtCu/C), 878 (for 1.0 V of PtCu/C) and around 180 (for Pt/C). Please explain why values are different and which value is correct.

In table 1, the current values 417 for PtCu/C (1.2 V), 878 for PtCu/C (1.0 V) and ~ 180 (for Pt/C) are the values of mass activity (Imass, A/g (Pt )). The specific currents (A/m2(Pt)) in Table 1 are now labeled Isp. These values are determined at a potential of 0.90 V. In Figure 5, the maximum currents of 170 and 150 A/g for PtCu/C and Pt/C, respectively, correspond to the limiting diffusion current Id, which is observed at potentials below 0.85 V. A detailed description of the method for calculating the kinetic current , by which Imass and Isp are defined, we added to the "Materials and Methods" section.

  1. Please analyze ORR stability in terms of chronoamperometry method. It is helpful for authors to confirm both ORR and MOR stability of PtCu/C.

The aim of our study is to study the effect of activation conditions on the activity of electrocatalysts in ORR and MOR. Assessment of the stability of catalysts involves long-term stress tests. It can be assumed that in this case the role of preprocessing will not be so significant. However, this is a different task. In our article, the emphasis is placed on the fundamental aspects associated with revealing the role of the PtCu structure reorganization of bimetallic nanoparticles under various conditions of their preliminary activation.

  1. Please explain why the particle size of PtCu/C is increased after 1.2V activation. Also, PtCu/C (1.2V) exhibits low ORR current compared with PtCu/C (1.0V). Please explain the reason and advantage of activation in potential range up to 1.2 V vs RHE in terms of “solid solution” structure.

Thank you for your comment. As part of the discussion of the results of TEM for the catalysts after activation, we gave a more detailed description of changes in the microstructure of materials: “We assume that a part of the PtCu/C nanoparticles of the sample after activation to 1.2 V also decreases in size due to selective dissolution of copper, as in the case of activation to 1.0 V. At the same time, the agglomeration of nanoparticles intensifies, leading to the formation of larger particles up to 11 nm (Fig. 6f). That leads to the preservation of the same average particle size as in the “as-prepared” sample. Thus, the average size of nanoparticles does not undergo significant changes as a result of activation, but the nature of the particles size distribution changes."  

TEM results indicate that the microstructure of catalysts activated under different conditions is somewhat different. The reasons for the significant difference in activity may be differences in the composition and structure of the surface layers of nanoparticles. Unfortunately, for nanoparticles of this size, it is difficult to confirm directly differences in the composition/structure of surface layers at the atomic level. At the same time, the effects of the surface reorganization influence of Pt-Cu nanoparticles caused by the treatment of catalysts in acids are described in [Gan, L .; Heggen, M .; O'Malley, R .; Theobald, B .; Strasser, P. Understanding and Controlling Nanoporosity Formation for Improving the Stability of Bimetallic Fuel Cell Catalysts. Nano Lett. 2013, 13, 1131-1138; Wang, D .; Yu, Y .; Zhu, J .; Liu, S .; Muller, D.A .; Abruña, H.D. (2015). Morphology and Activity Tuning of Cu3Pt/C Ordered Intermetallic Nanoparticles by Selective Electrochemical Dealloying. Nano Lett. 2015, 15, 1343-1348.], and revealed for Pt-Ni systems [https://doi.org/10.1021/nl304488q]. In making assumptions about the causes of the phenomena we observe, we are partly based on literature data.

  1. In Table 1, electron transfer number is higher than 4. I think that in error. Please analyze electron transfer number by using rotating ring disk electrode (RRDE) in order to calculate correct number.

Determination of the electrons number (n) is required only to confirm the 4-electronic ORR mechanism. In our work, as in many publications, the accuracy of determining n is no more than ± 0.2. Therefore, all the n values presented in Table 1 confirm the 4-electron mechanism of oxygen electroreduction, and the values 3.9 and 4.3, in fact, correspond to n = 4.

  1. In Table 1, cyclic voltametric activation up to 1.2 V vs RHE is much harsher condition than that of the 1.0 V. However, after activation, 1.0 V sample has 0.24 of Cu amount while 1.2 V sample has 0.25 of Cu amount. Please explain why 1.2 V sample has much copper amount.

As indicated in the "Results and Discussion" section, the compositions of the PtCu/C samples after two types of activation are close. The accuracy of determining the compositions of materials by XRF does not allow us to speak of a significant difference in the compositions of PtCu0.24 and PtCu0.25. Taking into account the remark, in the "Materials and Methods" section we indicate the error of the determination: “The error of the XRF method in determining the composition was PtCux±0.02.”

  1. We suggests that authors analyze BET surface area and compare with the ECSA results.

When determining the surface area of ​​PtCu/C materials by the BET method, the main contribution to the determined value will be made by the surface of the carbon support, which is about 270 m2/g. It is not possible to separate the fractions of the surface related to metal nanoparticles and carbon support. Moreover, the accuracy of the method is not very high. Judging by the results of more accurate electrochemical measurements, the ESA of the PtCu/C samples after different types of activation are very close: 41 and 43 m2/g(Pt). Therefore, we believe that for the systems under study, the method for assessing the surface area from low-temperature adsorption/desorption of nitrogen cannot provide significant information.

Round 2

Reviewer 2 Report

The manuscript evaluates the catalytic activity of PtCu1.3/C-alloy containing bimetallic nanoparticles. It is rather well written with adequate introduction, results and discussion. It is reported that bimetallic de-alloyed electrocatalysts are significantly more active than commercial Pt/C.

The title is unclear and must be corrected and improved as to express the presented results. The lines 400-402 need also to be reformulated. See for example: “maked it possible to obtain 400 a sample, its ORR activity being about 2 times higher than that of the sample activated in 401 the potential range from 0.04 to 1.2 V.”  

Author Response

The authors express their gratitude to the respected reviewers for carefully reading the manuscript, comments and recommendations made! The changes made are marked with a yellow marker. Below are the answers to the questions and comments of the reviewer.

The title is unclear and must be corrected and improved as to express the presented results.

agree, we changed the title to «Influence of Electrochemical Pretreatment Conditions of PtCu/C Alloy Electrocatalyst on its Activity».

The lines 400-402 need also to be reformulated. See for example: “maked it possible to obtain 400 a sample, its ORR activity being about 2 times higher than that of the sample activated in 401 the potential range from 0.04 to 1.2 V.” 

 We have reformulated this phrase

The voltammetric activation mode of PtCu1.3/C-alloy catalyst in the potential range from 0.04 to 1.0 V maked it possible to obtain a sample with an activity in ORR about 2 times higher than that of the sample activated in the potential range from 0.04 to 1.2 V.

Reviewer 4 Report

I recommend acceptance of the manuscript after following revision. Further review is not necessary.

1. Please show the graph including Ik values (417 A/g and 878 A/g).

2.  Please analyze XPS to confirm the electronic structure of PtCu/C.

Author Response

  1. Please show the graph including Ik values (417 A/g and 878 A/g).

These values are not shown in the graph, but were calculated using the Kutetsky-Levich equation.

A detailed description of the calculation is presented in the attached file.

  1. Please analyze XPS to confirm the electronic structure of PtCu/C.

Since after electrochemical activation, a very small amount of substance remains, which does not allow XPS measurements. In this case, it is impractical to measure only the PtCu1.3/C-alloy material prior to electrochemical activation, since the main interest is the change in the surface composition after activation in different conditions.
